# Increasing the Awareness of Under-Diagnosed Tropical Cases of Dengue in Romania

**DOI:** 10.3390/tropicalmed8100469

**Published:** 2023-10-07

**Authors:** Ioana Miriana Cismaru, Maria Adelina Radu, Ani Ioana Cotar, Florin Oancea, Violeta Melinte, Cristina Vacaroiu, Isabela Ghemulet, Valeriu Gheorghita

**Affiliations:** 1Faculty of Medicine, Carol Davila University of Medicine and Pharmacy, 020021 Bucharest, Romaniadr.melinte@gmail.com (V.M.); ctistina.vacaroiu@gmail.com (C.V.); 2Agrippa Ionescu Clinical Emergency Hospital, 011356 Bucharest, Romania; 3Cantacuzino National Institute of Research-Development for Microbiology and Immunology, 020021 Bucharest, Romania; cotar.ioana@cantacuzino.ro (A.I.C.);

**Keywords:** arbovirus, dengue fever, travel imported illness, imported febrile illness, tropical disease awareness

## Abstract

Global travelling increases every year and according to a report released during the COVID-19 pandemic by the UN World Tourism Organization, international travel doubled in 2022, compared to levels in 2021. his fact led also to travel-imported cases of arboviral infections and physicians are often confronted with tropical diseases, such as dengue or chikungunya. Since there is are no pathognomonic cues for these tropical illnesses, early diagnosis is still a big challenge and it depends on many factors, such as exposure risk factors, the epidemiological context, the incubation period, and the wide spectrum of differential diagnoses, including cosmopolitan or exotic infections. Since the clinical presentation of dengue is not typical and there are other febrile illnesses similar to arboviral diseases, misdiagnosis is common even among experienced doctors. Differential diagnosis needs up to date knowledge considering the short viraemic period, the antibody cross-reactivity, and the traps in recognising the nonspecific symptom picture. We present two cases of Dengue diagnosed in Romania which were initially clinically misconstrued, despite the characteristic symptom picture. The main purpose is to increase the level of awareness and to underline the difficulties that clinicians face in recognizing travel-related imported dengue virus disease.

## 1. Introduction

Dengue disease is caused by a mosquito-transmitted virus of the family *Flaviviridae*, principally *Aedes aegypti*, and represents the main cause of arthropod-borne viral disease globally [1,2,3]. The febrile illness is endemic in tropical regions worldwide, including Southeast Asia, South and Central America, Oceania, Africa, and eastern Mediterranean regions [4,5]. Symptoms are often non-specific; therefore, an accurate and efficient diagnosis of dengue is important for clinical care and surveillance support.

According to the last ECDC epidemiological report, this primary vector, which was not established before 2022 in the EU/EEA, was also confirmed in Cyprus and may continue to spread to other European Countries and outermost regions (e.g., Madeira, Martinique) due to global warming, travelling, climate changes and globalisation [6,7]. Apart from this main vector, to date, *Aedes albopictus* is the principal competent vector for the dengue virus in mainland EU/EEA and is largely established throughout the region [8].

The findings indicate that in 2023, until June, 2,162,214 cases of dengue were reported, of which 974 resulted in death, the majority in Brazil (387), with an important outbreak also in Argentina [9,10]. To date, no autochthonous case has been reported yet in Europe. Comparatively, in 2022, France reported 22 autochthonous cases and 6 cases in 2021 [11]. Although we can notice a worldwide trend towards modernisation and an increase in medical system standards which reduce the incidence of other tropical diseases, such as malaria, diseases such as Dengue are expanding their geographical range as an effect of intense traveling and globalisation [12].

In Romania, for 2023, the reference laboratory for diagnosing tropical diseases, the Cantacuzino National Institute of Research-Development for Microbiology and Immunology; Bucharest, Romania, 020021, received 50 laboratory sample requests for confirming dengue virus, out of which only 12 were confirmed by dengue-virus-specific IgM antibodies and 1 was confirmed by the presence of NS1 antigen; For 2022, there were 61 laboratory requests, out of which 20 were confirmed. 

There are four antigenically distinct serotypes, DENV1–4, called DENV-1, DENV-2, DENV-3, and DENV-4, and infection can be the result of any one or more than one of the four serotypes. Lifelong immunity can be the result of the infection by one serotype but the lifelong immune response does not cover by the other serotypes [13].

As the fastest spreading mosquito-borne viral disease worldwide and with major public health importance, Dengue is a re-emerging neglected tropical disease which has broad clinical spectra with an overlap of symptoms and signs that can easily be misdiagnosed [14,15]. According to their nonspecific nature, the clinical presentation for dengue infection might be characterised by fever, headache (particularly retroorbital pain), rash, myalgia, arthralgia, haemorrhagic manifestations (e.g., petechiae, bleeding gums, epistaxis, haematuria), weakness. Despite the WHO classification scheme including the above symptom picture, one study showed that its sensitivity depends on the patient’s age, decreasing inversely related to it, making detection of dengue tropical disease more difficult with older patients [16].

One challenge when diagnosing the dengue disease is the fact that symptoms appear after an incubation period of 5–7 days, up to 14 days after the initial contact. The long incubation period can pose challenges regarding accurate positive identification of the dengue virus considering the late onset of this pathology [17].

Symptoms may resemble these, but be related to a number of different diagnoses. Diagnosing the disease remains challenging due to the symptom picture that can mimic other medical conditions such as HIV, Epstein–Barr infection and Cytomegalovirus, Rubella, measles or parvovirus B19, which can also cause skin rashes, as well as bacterial infections, such as leptospirosis or syphilis with a classic palmar-plantar rash that appears in the secondary stage of this disease [18,19]. Even if there are similarities in the clinical presentations, there are differences which, together with the symptoms’ overall picture and epidemiological context, can overcome difficulties in obtaining the correct diagnosis and treatment [20].

Dengue laboratory diagnosis is essentially based on detection of the virus, with components or antibodies directed against the virus in blood samples. Direct diagnosis of dengue infection is based on virus isolation and detection of the viral genome by reverse transcription polymerase chain reaction (RT-PCR) or detection of NS1 antigen. Indirect diagnosis using serological methods to detect anti-DENV IgM and IgG is commonly employed, while IgA tests remain less commonly used. The selection of diagnostic methods depends greatly on the time-point of the sample collection during the course of the disease [21].

RT-PCR and virus isolation require a blood sample collected during the early febrile phase of the disease (0–5 days after the onset of fever) [22]. A sample obtained during the early phase is also preferred for NS1 detection, but in patients experiencing a primary infection, the NS1 antigen remains detectable for nine days or more after the onset of fever [23,24]. Serological methods can be used later during the course of the disease. IgM and IgA, however, can persist in blood for several weeks or even months after the infection, while IgG may persist for decades. Differentiating between an acute and a recent DENV infection by these methods can therefore be challenging.

We present two cases of Dengue diagnosed in Romania which were clinically misconstrued during the initial medical evaluation, despite the characteristic symptom picture. The main purpose is to increase the level of awareness to underline the difficulties that Romanian clinicians face in recognizing travel-related imported dengue virus disease. We also analyse similarities and differences in clinical and laboratory findings.

## 2. Detailed Case Description

### 2.1. Case 1

The first case report is a 33-year-old male Romanian patient, presented in May 2023 to the Agrippa Ionescu Clinical Emergency Hospital, in Bucharest, Romania. The epidemiological context for the patient is as follows: recent history of returning from Bali (12 days prior to presentation) without prophylaxis of tropical diseases, or any known allergies, who had the most common eruptive childhood diseases. He noticed a mosquito bite during travel through the Balinese tropical forest, he denied underwater activities, and confirmed that he consumed traditional Balinese street food. The symptoms began 12 days after his return to Romania, with malaise, a 40-degree fever, an intense frontal headache, retroorbital pain, a rash on the chest and arms, and low values of vital signs (arterial pressure = 80/50 mmHg). During the second day of illness he had the initial medical evaluation in the nearest local regional emergency room. The doctors raised the suspicion of an allergic syndrome, for which he was given hydrocortisone and subcutaneous adrenaline. No clinical improvement was noticed except for his blood pressure, so he asked to be discharged. 

During the fourth day of disease, 14 days after returning to Romania, he was admitted to the infectious diseases department, presenting local redness after a mosquito bite on the right leg, a positive tourniquet test, fever, generalized exanthema, ecchymoses after removing the suction ECG electrodes cup, bleeding nasal striae, bilateral subconjunctival haemorrhage, intense lower limb arthralgia, and a general rash with islands of skin sparing Figure 1.

At admission, a biological exam highlighted lymphocytopenia (0.9 × 10^3^,—reference range = 1.3–2.9 × 10^3^ mild monocities (1 × 10^3^—reference range = 0.3–0.8 × 10^3^) and thrombocytopenia (platelet count = 100 × 10^3/^μL—reference range = 150–400 × 10^3^), hepatic cytolysis syndrome with an alanine aminotransferase peak of 320 U/L (reference range 0–31 U/L), mildly elevated aspartate aminotransferase with a peak level of 112 U/L (reference range 0–31 U/L), lactate dehydrogenase with a peak of 202 U/L (reference range = 0–148 U/L ), minimal inflammatory syndrome with a C-reactive protein peak of 32 mg/L (reference range = 0–5 mg/L), negative blood cultures, a peripheral smear with no pathological major findings, and an increased INR value (1.6). A CT scan revealed a thin layer of pericardial fluid. Corroborating the epidemiological tropical context and the clinical and paraclinical laboratory findings, further diagnostic assessment was necessary and we raised the suspicion of viral febrile exanthema. We performed laboratory tests and imaging studies for infectious and non-infectious causes, such as dengue, Zika, chikungunya, Epstein–Barr virus infection, Cytomegalovirus infection, acute retroviral syndrome, rubella and measles, including differential diagnosis for endocarditis or haematological malignancy, e.g., acute myeloid leukaemia; asked for cardiology and haematology consultation and those were excluded.

Paired serology for Leptospirosis was negative. The laboratory diagnosis for this case was performed in the Laboratory for Vector-Borne Infections at the Cantacuzino National Institute of Research-Development for Microbiology and Immunology; Bucharest, Romania, 020021. For the first case, the following diagnostic tests were performed: detection of antibodies IgM and IgG against dengue, chikungunya and Zika viruses, using ELISA kits (Euroimmun, Berlin, Germany), as well as dengue NS1 antigen detection, using a dengue virus NS1 ELISA kit (Euroimmun, Berlin, Germany) according to the manufacturer’s protocol. The results obtained were negative for an IgG response to chikungunya virus, as well as for IgM and IgG responses to Zika virus. Positive results were obtained for both IgM and IgG reacting against dengue virus, as well as for IgM response to chikungunya virus. The IgM index to dengue virus was higher (8.87) compared with the IgM index to Chikungunya virus (2.54). The results obtained in the laboratory tests offered evidence for the confirmation of a dengue diagnosis.

The patient was administered a supportive hydration regime and acetaminophen. The outcome was favourable with a normalisation of blood count at the outpatient review and the disappearance of symptoms. The patient was discharged hemodynamically stable, with flaky skin lesions.

### 2.2. Case 2

A 62-year-old Romanian male who went to a private emergency hospital in June 2023, with a 7-day history of fever, fatigue, malaise, arthralgias and a nonpruritic generalized macular rash on his thorax, arms and legs, with purpuric lesions on his legs, as seen in Figure 2, starting 3 days after returning from a journey of 1 month in Thailand. From the patient history we found a percutaneous coronary intervention six months prior with double antiaggregant therapy continuing through to the present. It was labelled as a common cold with purpuric rash due to the antiaggregant therapy and the patient was referred home with symptomatic therapy and a recommendation of stopping the double antiplatelets therapy.

During the tenth day of illness, 12 days after returning to Romania, the patient presented at the Agrippa Ionescu Emergency Hospital., Bucharest, Romania, with persistent fatigue, malaise and a rash on his legs, without fever. On clinical examination, we noticed conjunctivitis and a nonpruritic macular rash on his thorax and abdomen with purpuric lesions on his legs, without other significant findings Figure 2.

Laboratory tests revealed mild lymphopenia (1 × 10^3^,—reference range = 1.3—2.9 × 10^3^), mild monocities (1 × 10^3^—reference range = 0.3–0.8 × 10^3^), thrombocytopenia (platelet count = 137 × 10^3^/U/L—reference range = 150–400 × 10^3^), and minimal inflammatory syndrome, hepatic cytolysis syndrome with an alanine aminotransferase peak of 280 U/L (reference range 0–31 U/L), mildly elevated aspartate aminotransferase with a peak level of 109 U/L (reference range 0–31 U/L), and a C-reactive protein peak of 12 mg/L (reference range = 0–5 mg/L). Blood cultures, urinalysis and bacterial and viral pharyngeal swab tests did not reveal any pathological findings. A hematologic and rheumatologic consult was performed in order to rule out vasculitis. Due to the epidemiological context, recent history of traveling to a high-risk area of tropical diseases, malaria was ruled out through a rapid diagnostic test and thin and thick blood film examination. We raised the suspicion of an acute arbovirus infection and sent blood samples to the reference laboratory of the National Institute for Medical-Military Research and Development “Cantacuzino” in Bucharest, where ELISA testing for dengue virus was positive for both IgM and IgG antibodies. The patient was discharged with symptomatic treatment and cardiac evaluation, with the recommendation of continuing the double antiaggregant therapy. At his one-week check-up, the rash had disappeared and the blood tests were normal. No other signs and symptoms appeared during the first month follow-up.

## 3. Discussion

As the fastest spreading mosquito-borne viral disease worldwide and with major public health importance, dengue is a re-emerging neglected tropical disease that has a broad clinical spectrum with an overlap of clinical manifestations and signs that can easily be misdiagnosed, due to similarities to other diseases caused by pathogenic microorganisms, such as bacteria, fungi, viruses etc. The overall symptom picture of this vector-borne disease can range from severe illness with a life-threatening course to asymptomatic. In light of a wide spectrum of differential diagnoses, it is necessary to consider the clinical similarities of signs and symptoms, which can be seen in Table 1, for a rapid and accurate diagnosis. Most arboviral diseases (dengue, Zika, chikungunya) are closely related and possess fever as a common feature, and often exhibit atypical exanthems [25]. Distinguishing viral exanthems from other life-threatening bacterial and rickettsia diseases with similar cutaneous manifestations may be crucial. Common laboratory characteristics for many viral infections, including arboviruses, are decreased lymphocytes and platelet counts [26], as can be seen in Table 2.

According to the WHO guidelines for differential diagnosis and initial formation, symptomatic dengue as a febrile disease, including rashes, muscle and joint pain, headache, and retro-orbital pain, might need additional objective findings, such as fluid accumulation, liver enlargement, a positive tourniquet test, leukopenia, and thrombocytopenia [26]. Differential discussions of symptom areas and different weighting of the same diagnostic information, depending on patient ethnicity, can lead to a differential diagnosis. The large diversity of arthropod-borne virus species identified in humans can provide a challenge in clinical diagnosis, due to the similarities in clinical manifestations, as shown in Figure 3. Another challenge is identifying the right aetiology for the febrile viral exanthem, which is hardly distinguishable from other viral infections. Additionally, one clinical diagnostic problem we should consider in Eastern Europe is the low incidence of non-autochthonous dengue cases [27,28].

Based on all the factors mentioned above, we consider that the two imported tropical cases of dengue can provide a good opportunity to understand this rare pathology in Romania. Considering our two cases, we observe the following characteristics: the first common issue that we noticed is the neglected tropical-epidemiological context of travel prior to hospitalisation in our medical department having relevance to the rare incidence and non-specific clinical presentation. This issue has also been acknowledged worldwide by the World Health Organisation (WHO), and was described in a public document entitled *A road for neglected tropical diseases 2021–2030*, which is the WHO’s second blueprint for preventing, controlling and, where feasible, eliminating and eradicating neglected tropical diseases. It follows the first roadmap, “Accelerating work to overcome the global impact of neglected tropical diseases”, issued in 2012, which set out global targets and milestones to be reached by 2020 for the 17 NTDs that comprised the WHO’s NTD portfolio at that time [29].

When considering the intensification of climate changes and the dynamics of mosquito borne-diseases affected by climate warming, we can refer to two studies following this relationship. One was carried out in 2019 using a mechanistic temperature-dependent transmission pattern and one was carried out in 2021. The first study showed that, by 2080, climate warming increases the risk of dengue outbreaks, with the largest population projected in Europe. The second study showed that 68% of the European continent will become favourable for the development of *Ae. albopictus* populations [30,31].

An important issue that we faced was differentiating dengue from other viral or bacterial diseases that presented similar symptoms, such as: fever, retro-orbital pain, malaise, asthenia, fatigue, chills, arthralgias, muscle aches, nausea/vomiting, and lymphadenopathy. The differential diagnosis excluded Epstein-Barr infection, cytomegalovirus, measles, rubella, retroviral syndrome, Zika and chikungunya virus, West-Nile virus, secondary syphilis, Listeria infection, Hepatitis, haematological disease, endocarditis.

The major challenge we managed regarding the first case we presented above was the late onset of symptoms correlated with the neglected tropical epidemiological context. Considering that the leading cause of viral febrile is not synonymous with dengue fever, the diagnostic key was to recognise the hypothesis of a tropical disease despite the long incubation period of 12 days (normal range of 3 to 10 days), then performing an accurate anamnesis complementary with the positive laboratory test for IgM Antibody Capture Enzyme-Linked Immunosorbent Assay. Due to the low incidence and the clinical similarity of this particular disease with other viral pathologies, clinicians are inclined to under-diagnose this particular pathology. This fact is supported by the presumptive diagnosis that the patient received during their first presentation to the medical unit—allergic disease.

The second case was diagnosed for the first time as a common cold with a purpuric rash associated with Clopidogrel, with an additional challenge—the patient’s age. Diagnosis of underlying causes of purpura in the elderly can be challenging, due to additional chronic comorbid conditions and pharmacologic treatment. Often it requires a multidisciplinary team including a haematologist, a dermatologist, and a rheumatologist. The differential diagnosis for purpura in elderly people without a known underlying condition should consider bleeding disorders (acquired haemophilia, acquired platelet disorders), autoimmune disorders (idiopathic thrombocytopenic purpura, cutaneous vasculitis), cancers, purpura associated with abnormal platelet dysfunction secondary to use of antiaggregant agents, or simple aging of the skin (senile purpura) [32].

In our case, the viral rash displayed purpuric lesions secondary to microbial endothelial damage and was intensified by pre-existing antiaggregant therapy. The presence of antiplatelet agents (clopidogrel and aspirin) led to misdiagnosis of the patient at his first presentation. Cases of thrombotic thrombocytopenic purpura (TTP) associated with clopidogrel were described in the literature. Many studies suggest that Clopidogrel-associated TTP is 15 times more likely to occur within the first 2 weeks of drug use [33]. The clues for certain diagnosis remained the history of travel in high-risk areas for tropical diseases and the presence of fever. Dengue disease presents a concerning public health problem for both the patient and community, with considerations such as immunizations and immunocompromised individuals. 

In clinical settings where diagnosis confirmation is promptly needed to guide management, rapid tests might be chosen for their speed and simplicity, in exchange for sensitivity and specificity. For instance, a rapid antigen test that detects a dengue-specific marker, such as DENV non-structural protein 1, a protein secreted from infected vertebrate cells, is now commercially available and is being used clinically [34]. However, despite the accuracy and availability of these laboratory tests, diagnosis of dengue would still first require recognition by the healthcare providers who would be ordering these tests.

Dengue virus (DENV) consists of four antigenically distinct serotypes (DENV-1, DENV-2, DENV-3, and DENV-4), which display high degrees of antigenic cross-reactivity with each other, as well as with other mosquito and tick-borne flaviviruses, such as Japanese encephalitis (JE), yellow fever, West Nile, and tick-borne encephalitis viruses [35]. The detection of the highly specific dengue NS1 antigen is possible in the serum of dengue virus-infected persons from the onset of clinical symptoms in primary and secondary infections [36]. Therefore, this antigen detection is an important tool for the diagnosis of acute dengue virus infections. 

A study performed in Romania and published in 2015 showed that Romanian tourists travelling to dengue-endemic countries are at risk of acquiring dengue infection [37]. In that study, serological and molecular diagnostics were performed on samples obtained between 2008 and 2013 from travellers with suspected dengue. The infections were acquired in endemic regions in Asia, Africa and in Europe (Madeira Island).

In 2022, the Laboratory for Vector-Borne Infections from “Cantacuzino” National Military Medical Institute for Research and Development tested samples collected from 50 suspected cases of dengue fever. They found that 12 of them were IgM/IgG positive and one case was NS1 antigen positive, whereas from the beginning of 2023 until present, samples from 61 suspected cases of dengue fever were tested and 20 positive results of IgM/IgG or NS1 antigen were obtained.

Therefore, mosquito-borne viral diseases are spreading worldwide, with a broad clinical spectrum, an overlap of clinical manifestation, and signs that can easily be misdiagnosed, representing a major public health issue.

## 4. Conclusions

Considering the potential for dengue disease to become fatal, misdiagnosis not only makes epidemiological understanding of the burden of affected diseases difficult, but also may have a deep effect on a patient’s health status by delaying supportive treatment with an increased probability of lethality.

Evaluating both cases and correlating the results with previous diagnosis, we understand the need to build on past lessons and experiences so that we can better grasp the emerging challenges regarding the current epidemiological status, the burden of disease, and clinical diagnosis. This will lead to a better preventive medical strategical intervention [37].

## Figures and Tables

**Figure 1 tropicalmed-08-00469-f001:**
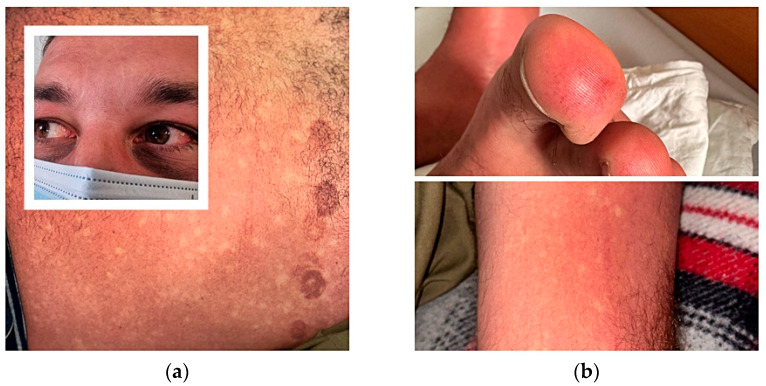
(**a**) Subconjunctival haemorrhage. Blanching rash with islands of skin sparing and ecchymoses after removing suction ECG electrodes cup; (**b**) Finger cutaneous vasculitis, positive tourniquet test.

**Figure 2 tropicalmed-08-00469-f002:**
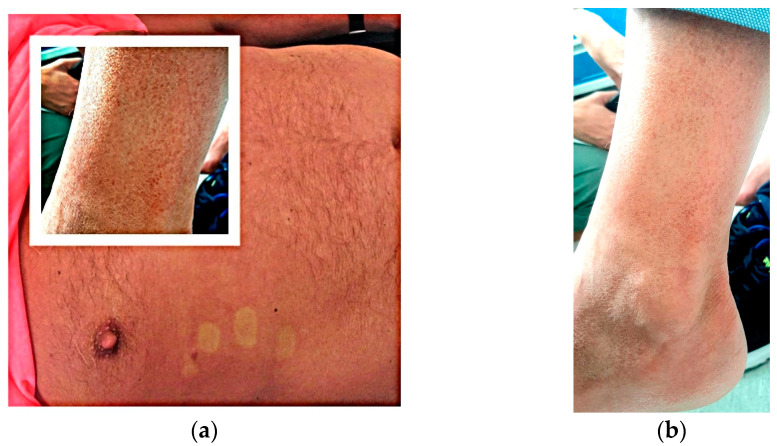
(**a**) Macular rash on thorax and purpuric lesions on legs; (**b**) Purpuric lesions on legs.

**Figure 3 tropicalmed-08-00469-f003:**
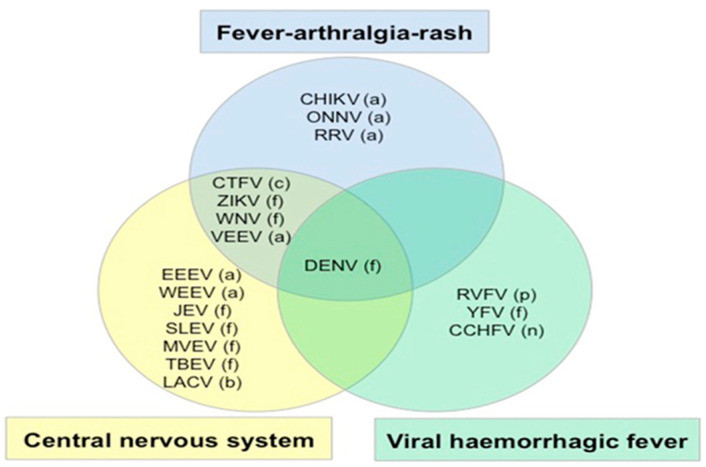
Summary of arbovirus syndromes together with fever: central nervous system, fever arthralgia rash and viral haemorrhagic fever. (a) alphavirus, (c) coltivirus, (f) flavivirus, (b) bunyavirus, (n) nairovirus and (p) phlebovirus. CCHF, Crimean Congo haemorrhagic fever; CHIKV, chikungunya; CTFV, Colorado tick fever; DEN, dengue; EEEV, Eastern equine encephalitis; JEV, Japanese encephalitis; LACV, La Crosse virus; MVEV, Murray Valley encephalitis; ONNV, O’nyong-nyong virus; RRV, Ross River fever; RVFV, Rift Valley fever; SLEV, St Louis encephalitis; TBEV, tick-borne encephalitis; VEEV, Venezuelan encephalitis; WEEV, Western equine encephalitis; WNV, West Nile fever; YFV, yellow fever; ZIKV, Zika virus. Adapted with permission from Solomon T, chapter 40 in Beeching N, Gill G, eds., Lecture notes: tropical medicine (Wiley: New York, NY, USA; 2014), p. 274.

**Table 1 tropicalmed-08-00469-t001:** Comparison of selected clinical findings of dengue infection in the two cases.

Clinical Presentation	Case 1	Case 2
Fever	+++	+
Maculo-papular exanthema	+++	++
Myalgia	++	+
Arthralgia	+++	+
Oedema	+	-
Retro-orbital pain	++	-
Conjunctivitis	++	-
Lymphadenopathy	-	-
Hepatomegaly	-	-
Haemorrhage	+	-
Incubation period (define by days between return from the epidemiological area and onset of symptoms)	12	3

+++ very common; ++ frequently observed; + sometimes observed; - not typical.

**Table 2 tropicalmed-08-00469-t002:** Comparison of baseline laboratory findings of dengue infections in the two cases.

Laboratory Findings	Case 1	Case 2	Reference Values
Anaemia	12.7 g/dL	13 g/dL	(13–15)
Leukopenia	2.9 × 10^3^/μL	3 × 10^3^/μL	(4–10 × 10^3^)
Neutropenia	2.2 × 10^3^/μL	1.8 × 10^3^/μL	(2.2–4.8 × 10^3^)
Lymphocytopenia	0.9 × 10^3^/μL	1 × 10^3^/μL	(1.3–2.9 × 10^3^)
Thrombocytopenia	100 × 10^3^/μL	137 × 10^3^/μL	(150–400 × 10^3^)
Increased CRP	32 mg/L	12 mg/L	(0–5)
Increased ALT	320 U/L	280 U/L	(0–31)
Haematocrit	40%	43%	(35–47)

ALT, alanine aminotransferase; CRP, C-reactive protein

## Data Availability

No new data were created or analyzed in this study. Data sharing is not applicable to this article.

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
