# Peer review of "Increasing the Awareness of Under-Diagnosed Tropical Cases of Dengue in Romania"

_tropicalmed, 2023, doi:10.3390/tropicalmed8100469_

Round 1

Reviewer 1 Report

Estimated Authors,

I've read with great interest the present case report on two cases of travel-associated Dengue from Romania. According to ECDC official estimates (https://atlas.ecdc.europa.eu/public/index.aspx) no case of Dengue has been officially reported in Romania during 2021, compared to 15 in 2019, and 3 in 2020. At the time of this review, no estimates have been provided for 2022 and 2023. Therefore, the rationale for providing a case report instead of a more extensive case series is properly fulfilled: Dengue remains a rare diagnosis in Romania, at least compared to other EU/EEA countries, and providing details explaining why competent physicians may have failed to properly recognize the infection is very important for all potential readers.

The present report mostly achieve its goals, but some improvements are still required.

To begin with, the discussion section could benefit from some editing aimed to shorten it as much as possible without impairing its comprehensiveness. 

Second, Table 2 is useful but could be improved by reporting crude values with a further column with reference values. In this regard, please double check all reference values in the main text, as some have been removed during the editing of the paper (I guess).

Third, I would suggest to design a summary table 3 where Authors could summarize the differential diagnoses that have been initially considered for Pat. 1 and Pat. 2.

Fourth, could you provide the serotypes of diagnosed DENV?

The English is grammarly correct, but it could benefit from some editing on its flow.

Author Response

Respons to first Reviewer:

Estimated Reviewer,

First of all, we thank you for sharing your interest in our paper and also for all the sugestions. It is true that very few cases of Dengue fever were reported lately in Romania, due to no evidence of specific species of mosquitos circulation. Our paper is aimed to increase the awarness  of frequent imported cases from travelers to high-risk areas, especially during the summer period, from June to September, when most of the romanian people go to vacation in exotic countries. Unfortunatelly, we cannot provide the serotypes for Dengue viruses, but due to increasing cases , our refference laboratory is making efforts to be able to perform also the serotypes.

Reviewer 2 Report

1. This Old problem is causing misdiagnosis for younger physicians in Latin America, where the three arboviruses circulate concurrently in many regions. 

2. I like this manuscript because it alerts European countries about the coming of emerging vector-borne diseases.  The paper emphasizes three factors that are combined in the region: increase of poverty-infectious diseases-human migration. 

3. My feeling is that Discusion section has to be used used to educate many MD´s in Europe about a better NTD diagnosis in the First Level of Hospitals Attention.  Simply reasoning such as the lack of a right diagnosis in this level will drive to lack of laboratory testing, poor vector control practices and underdiagnosed epidemiological rates.  And the worse fact: budget cuts for NTD prevention-

3. I recommend to Authors to make comments on the need to diseminate this scientific findings in Traveler Medicine meetings, and why not to estimulate a european observatory for NTD´s.

4. I´d like to see some data on Aedes aegypti and Aedes albopictus distribution in Romenia and neighbouring countries. Italy has been publishing data of these two species.  Isolated buy consistent WNV cases in horses, humans, and birds are documented in the balcanic and mediterranean contries. 

5. Tick arboviruses traditionally are more frequently reported than mosquito-borne diseases in Europe.  A comments about it could assist MD´s to focus also in these vectors.

6. West Nile Virus has many factors to develop its cycle and potential outbreaks in the warmer countries along West Europe and closer asian countries. City crow populations are frequent during day time.  Culex and other Culicidae species prevails as well.  An important bird migratory path from Africa to Europe is well known.  It has to be a concerning potential of this VBD. 

Author Response

Respons to second Reviewer:

Estimated Reviewer,

First of all, we thank you for sharing your interest in our paper and also for all the sugestions. It is true that very few cases of Dengue fever were reported lately in Romania, due to no evidence of specific species of mosquitos circulation and missdiagnosis. Our paper is aimed to increase the awarness  of frequent imported cases from travelers to high-risk areas, especially during the summer period, from June to September, when most of the romanian people go to vacation in exotic countries. In order to increase the awareness of emerging vector-borne diseases we present clinical cases in many local conferrences, travel warning ads in which we encourage travelers to have a travel medicine consultation prior the journey. Unfortunately, we do not have data of any Aedes aegypti or albopictus species circulating in Romania, so all the Dengue cases are considered as been importated. Instead, indeed we have more frequent reported cases of west Nyle virus infection. So, due to bird migration path, increasing travelers to exotic areas, it may by possible to see in the near future a concerning potential of vectorn-borne diseases and increase of poverty-infectious diseases-human migration.